# Acid–Base Equilibrium and Self-Association in Relation to High Antitumor Activity of Selected Unsymmetrical Bisacridines Established by Extensive Chemometric Analysis

**DOI:** 10.3390/molecules27133995

**Published:** 2022-06-21

**Authors:** Michał Kosno, Tomasz Laskowski, Joanna E. Frackowiak, Agnieszka Potęga, Agnieszka Kurdyn, Witold Andrałojć, Julia Borzyszkowska-Bukowska, Katarzyna Szwarc-Karabyka, Zofia Mazerska

**Affiliations:** 1Department of Pharmaceutical Technology and Biochemistry and BioMedTech Centre, Faculty of Chemistry, Gdańsk University of Technology, Gabriela Narutowicza Str. 11/12, 80-233 Gdańsk, Poland; michal.kosno@pg.edu.pl (M.K.); joanna.frackowiak@pg.edu.pl (J.E.F.); agnieszka.potega@pg.edu.pl (A.P.); agnieszka.kurdyn@pg.edu.pl (A.K.); julbukow@pg.edu.pl (J.B.-B.); 2Institute of Bioorganic Chemistry, Polish Academy of Sciences, Zygmunta Noskowskiego Str. 12/14, 61-704 Poznań, Poland; wandralojc@ibch.poznan.pl; 3Nuclear Magnetic Resonance Laboratory, Gdańsk University of Technology, Gabriela Narutowicza Str. 11/12, 80-233 Gdańsk, Poland; katszwar@pg.edu.pl

**Keywords:** anticancer, unsymmetrical bisacridine, physicochemical properties, self-association, pK_a_ determination, chemometric analysis of spectra, UV-Vis spectroscopy, NMR spectroscopy, cancer, carcinoma

## Abstract

Unsymmetrical bisacridines (UAs) represent a novel class of anticancer agents previously synthesized by our group. Our recent studies have demonstrated their high antitumor potential against multiple cancer cell lines and human tumor xenografts in nude mice. At the cellular level, these compounds affected 3D cancer spheroid growth and their cellular uptake was selectively modulated by quantum dots. UAs were shown to undergo metabolic transformations in vitro and in tumor cells. However, the physicochemical properties of UAs, which could possibly affect their interactions with molecular targets, remain unknown. Therefore, we selected four highly active UAs for the assessment of physicochemical parameters under various pH conditions. We determined the compounds’ pK_a_ dissociation constants as well as their potential to self-associate. Both parameters were determined by detailed and complex chemometric analysis of UV-Vis spectra supported by nuclear magnetic resonance (NMR) spectroscopy. The obtained results indicate that general molecular properties of UAs in aqueous media, including their protonation state, self-association ratio, and solubility, are strongly pH-dependent, particularly in the physiological pH range of 6 to 8. In conclusion, we describe the detailed physicochemical characteristics of UAs, which might contribute to their selectivity towards tumour cells as opposed to their effect on normal cells.

## 1. Introduction

Unsymmetrical bisacridines (UAs) are a new class of compounds developed by our research group; they are characterized by a unique structure consisting of two heterocyclic ring systems linked by an aminoalkyl chain [1,2]. UAs exhibited high cytotoxic activity against all fourteen tested cancer cell lines as well as antitumor activity against Walker 256 rat adenocarcinoma and ten human tumor xenografts in nude mice. Notably, the compounds which displayed the highest activity strongly inhibited pancreatic cancer cell lines [3]. Studies on the biological effects of these compounds demonstrated their ability to suppress 3D cancer spheroid growth [4]. Moreover, their anticancer activity was enhanced when bound to quaternary quantum dots, resulting in selective upregulation of their cellular uptake [5,6,7]. Our previous studies demonstrated that UAs undergo metabolic transformations which retain the compounds’ dimeric structure. These include products of phase I and phase II metabolism such as N-oxide derivatives in the aminoalkyl side chain through flavin monooxygenase (FMO) transformations as well as uridine UDP-glucuronosyltransferase (UGT)-mediated glucuronide conjugates at the hydroxyl group in the imidazoacridinone ring, respectively [8]. We identified products resulting from glutathione S-transferase-mediated glutathione conjugation [9].

Our recent studies indicated that certain structural characteristics of UA dimers (Figure 1) which distinguish them from acridine monomers (Figure 2) might reflect their unique physicochemical properties under physiological conditions [10,11,12]. Although dimer compounds indeed consist of an imidazoacridinone and 1-nitroacridine monomer connected by a flexible linker, their properties are by no means a simple sum of characteristics displayed by monomeric compounds. For instance, acridine monomers have been shown to directly interact with DNA, which was confirmed by up to a 20-degree increase in the delta T_m_ of the double-stranded DNA helix [13,14]. However, our preliminary studies revealed that UAs increased this parameter by merely 5–7 degrees, which suggests non-specific interactions between UAs and the DNA helix [1]. Instead, it has been hypothesized that the unique dimer structure of UAs may promote stabilization of DNA quadruplexes [1]. However, due to the large structural variability of G-quadruplexes [15,16,17,18], it is difficult to determine the nature of these interactions.

In this context, it is essential to investigate how pH values relevant to normal and tumor tissue might affect the physicochemical properties of these anticancer compounds. Healthy tissue maintains a pH between 7.2 and 7.4 in the intracellular and extracellular space. However, while the intracellular pH of cancer cells remains approximately 7.4, the pH of their extracellular microenvironment is decidedly more acidic and can vary from 5.5 to 7.0 [19,20,21]. This suggests significance of a lower pH in relation to the physicochemical properties of UAs, and therefore their possible selectivity towards cancer cells.

In our preliminary studies, we identified at least two distinct pK_a_ values for UAs and their three different pH-dependent different forms [8]. The compounds’ protonation state would therefore affect their solubility in water and ultimately the nature of their possible interactions with various cellular components [22,23,24]. In particular, DNA G-quadruplexes may be the main target of anticancer UAs owing to negatively charged groups present in nucleic acids. Considering the above, it is crucial to fully understand variations in charge between pH-dependent structures of UAs, especially considering their potential to interact with DNA [25,26,27].

In the present work, we aimed to establish structures of all possible pH-dependent ionic forms of UAs, particularly at a pH relevant for tumor cells as compared to normal cells. In order to identify possible structural elements of UAs which might be crucial for interactions with DNA, we performed extensive chemometric analysis of UV-Vis spectra of four UAs selected on the basis of their high antitumor activity. The obtained data were supported by nuclear magnetic resonance (NMR) experiments and by the interpretation of TOCSY and ROESY 2D NMR spectra. This allowed us to establish protonation states and assess the aggregation propensity of the studied compounds (C-2028, C-2041, C-2045, and C-2053), the structures of which are shown in Figure 1. The purpose of the study was to gain insight into the relevance of physicochemical properties displayed by UAs in the context of their previously observed significant antitumor effect.

## 2. Results

### 2.1. Determination of pK_a_ Values for Studied Compounds

pK_a_ values for the studied unsymmetrical bisacridines were determined by analysis of their UV-Vis spectra under several pH conditions. It was found that the differences between the spectra of individual forms are rather inconsiderable. Therefore, a complex chemometric analysis of the obtained spectra was performed following the procedure described in Materials and Methods. Figure 3 depicts molar fractions of individual spectral forms for the studied UAs. The points represent the coordinates of the observations, which were calculated from experimental data using PCA. Solid curves depict the best fit of the theoretical model to the experimental data according to Equations (1)–(3), provided in Section 4.5.1.

Figure 3 presents the determined molar fractions of five independent spectral forms in relation to pH solution for C-2028, C-2045, and C-2053, and four for C-2041. The initial pK_a_ values were identified as points at the intersection of relevant molar fractions and were then optimized by simplex optimization. The obtained pK_a_ values and their standard deviations (SDs) were evaluated by leave-one-out cross validation and are summarized in Table 1.

A possible model of successive molecule deprotonation events in the studied pH range from 1.5 to 11 was proposed as described below. First, it was crucial to determine the pH-dependent structures of UA monomer analogues under the studied conditions. In this respect, pH-dependent forms of monomers were established using combined UV-Vis and chemometric methodology as well as advanced NMR spectroscopy. The structures of three analysed compounds: **5**. 1-alkylamino-8-hydroxyimidazoacridinone (Symadex^®^, C-1311; Xanthus Pharmaceuticals, Cambridge, MA, United States), **6**. 1-alkylaminoimidazoacridinone, and **7**. 1-nitro-9-alkylaminoacridine (Ledakrin, C-283), representing monomer units of UAs, are presented in Figure 2. Analysis of compounds (**5**) and (**6**) provided information regarding dissociation constants associated with the imidazoacridinone ring and the hydroxyl group substituted at position 8 in this ring, whereas compound (**7**), with the 1-nitroacridine ring. pK_a_ values determined for the described monomers, is displayed in Table 2.

These results allowed us to propose a sequence of proton dissociation for UAs. Analogous to the pK_a_ of phenol, the pK_a3_ (9.675) is most likely associated with the hydroxyl group in the imidazoacridinone ring; therefore, it is expected that the pK_a5_ value for the dimeric C-2045 represents the hydroxyl group. Moreover, the structure of C-2045 and C-2053 is identical apart from the hydroxyl group at position 8 of the imidazoacridinone ring in C-2045. The highest pK_a_ value determined for C-2053 was 8.235, suggesting that pK_a5_ is related to the hydroxyl group present in C-2045.

In the present work, we aimed to establish the structures of all possible pH-dependent ionic forms of UAs, particularly at a pH relevant for tumor cells as compared to normal cells. In order to identify possible structural elements of UAs which might be crucial for interactions with DNA, we performed extensive chemometric analysis of UV-Vis spectra of four UAs selected on the basis of their high antitumor activity. The obtained data were supported by nuclear magnetic resonance (NMR) experiments and by the interpretation of TOCSY and ROESY 2D NMR spectra. This allowed us to establish protonation states and assess the aggregation propensity of the studied compounds (C-2028, C-2041, C-2045, and C-2053), the structures of which are shown in Figure 1. The purpose of the study was to gain insight into the relevance of physicochemical properties displayed by UAs in the context of their previously observed significant antitumor effect.

### 2.2. UA Structure Determination by NMR Spectroscopy

The obtained pK_a_ results, which are summarized in Table 1, revealed the total number of protonation forms possible for each studied UA compound. Although comparative UV-Vis spectra analyses of UAs and their monomeric analogues provided a general idea of pH-dependent proton dissociation sequences, in certain cases additional experiments were required in order to paint the full picture. The missing dots were connected upon NMR studies of UAs at four selected pH values. The lowest and highest pH conditions were chosen to study the fully protonated and fully deprotonated UA species, respectively.

In order to enable spectroscopic observations of labile protons of UA molecules, the relevant NMR experiments, i.e., one-dimensional and 2D TOCSY and ROESY spectra, were recorded at 5 °C. At these conditions, the exchange ratio of amino, imino, and hydroxyl protons was significantly lowered due to the increasing proton-dissociation energetic barrier. Unfortunately, we were unable to record standard NOESY experiments for UAs, as these compounds exhibit molecular masses just below 1 kDa, which corresponds to the phase-shift region of the nuclear Overhauser effect. 

Our ^1^H NMR analysis of UAs (Figure 4) at pH~1 revealed signals (1) in the 11–12 ppm region and (2) in the 9.5–11 ppm region, both of which disappeared at pH = 7. These protons were identified upon unambiguous correlations, observed in ROESY and TOCSY spectra (data not shown), as: (1) an imino proton bound to the nitrogen atom of the 1-nitroacridine ring system and (2) an amino proton of the tertiary nitrogen in the linking chain, respectively. Hence, pK_a2_ values of 6.139, 5.843, 6.120, and 5.997 for C-2028, C-2041, C-2045, and C-2053, respectively, were assigned to the nitrogen atom of the 1-nitroacridine ring. Meanwhile, the amino group at position 9 of the 1-nitro-9-aminoacridine ring is not protonated and exists mainly as an imine at a pH lower than the pK_a_ value of the aromatic acridine nitrogen. Dissociation of this proton results in a negative charge of the entire aromatic system, which in consequence causes the imine group at position 9 to be more susceptible to hydrolysis (data not shown).

Continuing the above, the next pK_a3_ values, ranging from 7.158 to 7.522, were related to the proton dissociating from the tertiary nitrogen atom in the linking chain (previously referred to as point 2), whereas pK_a4_ values equal to 8.235 and 8.546 were assigned to the nitrogen atom in the imidazole part of the imidazoacridinone ring. It should be noted that the proton attached to the imidazole ring was not visible in the NMR spectra. These results demonstrate that the higher pK_a4_ values for C-2028 and C-2053, rather than being associated with the tertiary amino group of the side chain, are actually related to the nitrogen atom in the imidazole part of the imidazoacridinone ring. Moreover, pK_a5_ equal to 9.137 should relate to dissociation of the proton of the 8-hydroxyl group of C-2045, which is in line with the standard pK_a_ values of phenol-type molecules. A summary of this scheme is presented in Figure 5, illustrating the sequence of proton dissociation in UAs described above.

### 2.3. Self-Association of UAs

Several hundred UA spectra collected in selected pH and temperature conditions were used to study the self-association of UAs. Subsequently, chemometric analysis of the obtained spectra was performed according to the procedure described in Materials and Methods. Figure 6 depicts molar fractions of all studied compounds at pH 2.5. Points corresponding to molar fractions of independent spectral forms were determined on the basis of experimental data and were derived from coordinates of the observations obtained via PCA. Solid curves represent the best fit of the theoretical model to the experimental data.

The theoretical model of self-association was determined by numerical fitting. Three different aggregation models were considered: dimerization, unlimited aggregation, and micellar aggregation. The obtained self-association constants were evaluated via leave-one-out cross validation.

Figure 6 shows that at 35 °C and pH 2.5 aggregation proceeds similarly for C-2028, C-2045, and C-2053, while self-association of C-2041 is observed to a lesser extent. In this case, dimerization is the best model for all UAs. The results slightly differ for pH = 3.5 (Figure 7); while self-association remains a one-step process, the determined molar fractions no longer fit a simple aggregation model and unlimited self-association seems to be the most suitable model.

The cross-points of the individual molar fractions shifted towards lower concentration values, suggesting that the data fit a different self-association model, while the self-association constants clearly have lower values.

Moreover, in Figure 8 we demonstrate that pH equal to or greater than 6.5 increases the complexity of UA self-association, as it progresses in at least two stages. Aggregation of C-2041 at pH 6.5 is relatively less complicated than for the remaining compounds, as it occurs in a single step. While self-association analysis was additionally carried out at 4 °C, it failed to provide a consistent aggregation model due to the strong propensity for aggregation based on two forms. This was supported by elevated values of the self-association constant. All determined self-association constants including their standard deviations are listed in Table 3. These findings support the view that self-association of UAs is strongly pH-dependent, as it is directly correlated with increasing pH values.

## 3. Discussion

With respect to the aim of this study, our findings outlined in Table 1 indicate that the four UA antitumor derivatives (Figure 1) selected for this study exist as five (for C-2028, C-2041, and C-2053) or six (for C-2045) independent forms in the pH range from 1.5 to 11. Notably, at physiological pH (between 6 and 8) there are three to four individual protonation forms, each most likely capable of distinct interactions with DNA and therefore of diversely affecting cellular processes at the molecular level. Moreover, our studies were prompted by the current understanding that the pH value maintained in the extracellular environment of cancer cells is lower (5.5 to 7.0) than that in healthy tissue (7.2 to 7.4). This suggests that certain protonation states of the studied compounds might be selective towards cancer cells.

It was challenging to assign a particular pK_a_ value acquired by UV-Vis analysis, to the respective (de)protonating atom using of NMR. We were able to obtain preliminary assignments through an approach based on UA structural analogues. NMR analysis of the monomers Symadex^®^ (C-1311), Ledakrin (C-283), and 1-alkylaminoimidazoacridinone (Figure 3) provided evidence that pK_a_ values of approximately 2 and 9 were associated with the imidazoacridione ring, whereas 6 was related to the 1-nitroacridine ring.

NMR analysis confirmed the validity of the preliminary pK_a_ assignments obtained through analysis of UA structural analogues. Although detection of all exchangeable protons by NMR was not possible due to different exchange ratios with solvent molecules, this technique allowed us to recognize exchangeable protons related to the alkyl part of UAs. Moreover, NMR enabled us to determine the sequence of proton detachment. Our approach, incorporating the several strategies described above, allowed us to clearly define the exact structures of the individual UA protonation states, which are presented in Figure 5.

Furthermore, we were able to demonstrate that the investigated unsymmetrical bisacridines are prone to self-association. UV-Vis spectra extensively analysed by chemometric techniques revealed that all studied UAs underwent aggregation, which was strongly pH-dependent. Principal component analysis demonstrated the presence of at least two sources of variability in the spectra of various UA concentrations collected at several pH values. Based on the obtained results, we propose dimerization as the best fitted model of self-association in acidic media. Determining the nature of the aggregation processes at physiological pH proved to be more challenging. Presumably, it involves a dimer as the first order aggregate followed by a second order aggregate of greater complexity. The complicated nature of the aggregation process may result from distinct structural forms related to the described protonation states.

In conclusion, the results presented here can provide an insightful background for further studies including metabolic transformations of UAs and their possible interactions with nucleic acids and/or proteins considering pH differences between normal and tumor tissues. Proton exchange significantly rearranges the distribution of electron density in a compound, which in turn strongly influences its ability to self-associate, potentially impacting the anticancer activity of these compounds.

## 4. Materials and Methods

### 4.1. Chemicals

Unsymmetrical bisacridines: 9′-{*N*-[(imidazo [4,5,1-de]-acridin-6-on-5-yl)aminopropyl]-*N*-methylaminopropylamino}-10-nitroacridine (C-2028, **1**); 1-[3-(imidazo [4,5,1-de]-acridin-6-on-5-yl)aminopropyl]-4-[3′-(10-nitroacridin-1-yl)-aminopropyl]piperazine (C-2041, **2**); 9′-{*N*-[(8-hydroxyimidazo [4,5,1-de]-acridin-6-on-5-yl)aminopropyl]-*N*-methylaminopropylamino}-40-methyl-10-nitroacridine (C-2045, **3**); and 9′-{*N*-[(imidazo [4,5,1-de]-acridin-6-on-5-yl)aminopropyl]-*N*-methylaminopropylamino}-40-methyl-10-nitroacridine (C-2053, **4**) were synthesized and purified in our laboratory according to the method described in the relevant patents. HEPES and PIPES buffers, potassium cacodylate, sodium acetate, sodium hydroxide, and sodium tetraborate were purchased from Merck (Merck Millipore, Darmstadt, Germany). Deionized water (conductivity 0.056 μS·cm^−1^, Milli-Q water purification system; Merck Millipore) was used in all experiments.

### 4.2. General Procedures

All solvents and chemicals used required no further purification. Stock UA solutions (10^−2^ M) were prepared in deionized water and then diluted to a 10^−3^ M or 10^−4^ M concentration if necessary.

### 4.3. pK_a_ Determination

Hydrochloric acid or buffer (2 × 10^−1^ M) solutions of UAs (2 × 10^−5^ M) were prepared with the following pH values: 1.5, 2, 2.5, 3, 3.5, 4, 4.5, 5, 5.5, 6.1, 6.3, 6.5, 6.7, 6.9, 7.1, 7.3, 7.5, 7.7, 7.9, 8.1, 9, 10 and 11. Based on their buffering capacity, we used acetate, PIPES, HEPES, and borate buffer in the pH range from 4 to 11 and hydrochloric acid in the pH range from 1.5 to 3.5. UV-Vis spectra of the samples were recorded in a quartz cuvette with an optical path length of 1 cm.

### 4.4. Aggregation Study

UAs in the following concentrations: 10^−6^, 2 × 10^−6^, 5 × 10^−6^, 10^−5^, 2 × 10^−5^, 5 × 10^−5^, 10^−4^, 2 × 10^−4^, 5 × 10^−4^, and 10^−3^ M were prepared as HEPES buffer solutions (10^−1^ M) with pH values of 2.5, 6.5, 7.4, and 8.5. We investigated the propensity of UAs to undergo aggregation under the conditions described above at a temperature of 37 °C, as required for enzymatic transformations of UAs. UV-Vis spectra were collected at various temperatures (4 °C, 22 °C, and 37 °C) with the optical path length of a quartz cuvette varying from 0.1 cm to 1 cm.

### 4.5. Chemometric Analysis: General Procedures

All absorbance values were expressed as molar extinction coefficients. Spectral data were subjected to appropriate pre-processing involving correction of signal distortion and elimination of outliers. Subsequently, data were organized into a matrix and subjected to numerical decomposition into eigenvectors (Principal Component Analysis, PCA). Significant principal components were determined through a linear map and residual spectral analysis. Based on the retained number of significant principal components (n), PCA was repeated to extract the (n − 1) dimensional hyperplane from the n-dimensional space. For hyperplanes with dimensions greater than or equal to three, the coordinates of the observations (V) were set in an n-dimensional simplex. The corners of these entities were manually determined and then optimised using the Nelder–Mead method. The contributions of the principal components were assessed using the V and then optimized using the penalty function. Molar fractions were obtained based on the contributions of individual principal components and calculated using the coefficients of the plane equations. A relevant plane was identified using the linear regression method. For two-dimensional space, the initial and final points along the principal component were identified using the Nelder–Mead method. All steps of the performed chemometric analysis are described in detail in the Appendix A.

#### 4.5.1. Chemometric Analysis for pK_a_ Determination

Dissociation constants (pK_a_) were determined based on molar fractions using an extended version of the Henderson–Hasselbalch equation (Equations (1)–(3)):(1)$$\gamma =\frac{\prod_{i=1}^{4}K_{i}}{\prod_{i=1}^{4}H^{i}}$$
(2)$$u_{1}=\frac{1}{1+\gamma }\;$$
(3)$$u_{n}=u_{1}\cdot \gamma \cdot n=\left\{1,4\right\}\;$$

*K*—dissociation constant,

*u_n_*—molar fractions.

The initial pK_a_ values were identified as intersection points of the relevant molar fractions and then optimised using the Nelder–Mead method [28] (consult Appendix A for more details). The accuracy of the calculated molar fractions was estimated using leave-one-out cross-validation (LOOCV). This sophisticated methodology was employed to determine the pK_a_ values of multiprotic compounds in the 2–11 pH range.

#### 4.5.2. Chemometric Analysis for Self-Association Study

Self-association constants (K), for two-dimensional space, were identified by fitting the determined molar fractions into an appropriate theoretical model (dimerization), then adjusted using simplex optimisation. For three-dimensional space, K values were determined as reciprocals of the logarithm of concentration corresponding to the intersection of the molar fractions. All calculations were performed in the R environment (R Foundation for Statistical Computing, version 4.0.2, Vienna, Austria).

### 4.6. NMR Studies of UA Protonation Forms

In order to track the resonances of the labile protons of the studied UAs, relevant solutions (10^−3^ M) were prepared in the following buffers or acids (10^−2^ M): hydrochloric or sulphuric acid (pH = 1), cacodylate buffer (pH values of 5, 7 or 8), and borate buffer (pH = 11). Solutions contained 10% D_2_O, while those analysed by TOCSY and ROESY were prepared in pure D_2_O. Subsequently, sets of 1D and 2D NMR (TOCSY and ROESY) were recorded by a 700 MHz Avance III spectrometer (Bruker) equipped with a QCl CryoProbe (in collaboration with the Institute of Bioorganic Chemistry, Polish Academy of Sciences, Poznań, Poland). All spectra were processed using TopSpin software (Bruker GmbH, version 4.1.3, Karlsruhe, Germany). The TOCSY mix time was set to 80 ms, whereas the ROESY mix time was set to 300 ms.

Furthermore, a complete assignment of the non-exchangeable protons of the UAs was performed on the basis of an additional set of DQF-COSY, TOCSY, and ROESY spectra. These experiments were recorded at pH = 4.5 (acetate buffer) and a temperature of 25 °C using samples of compounds **1**–**4** at concentrations of 10^−3^ M dissolved in pure D_2_O. The TOCSY mix time was set to 80 ms, whereas the ROESY mix time was set to 300 ms. The resulting assignments as well as fragments of selected 2D NMR spectra are included in the Appendix A (Appendix A, Appendix A).

## Figures and Tables

**Figure 1 molecules-27-03995-f001:**
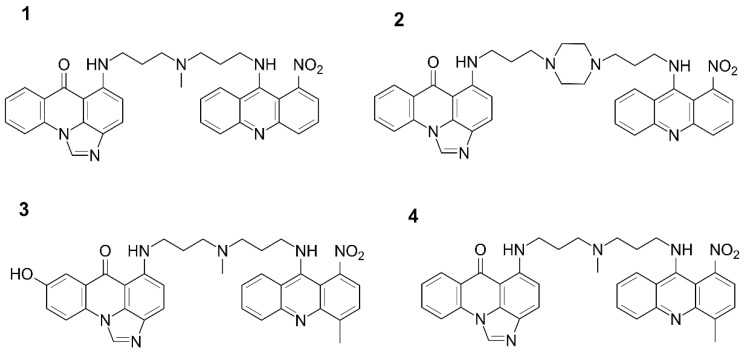
Chemical structures of the studied unsymmetrical bisacridines (UAs): C-2028 (**1**), C-2041 (**2**), C-2045 (**3**), and C-2053 (**4**).

**Figure 2 molecules-27-03995-f002:**
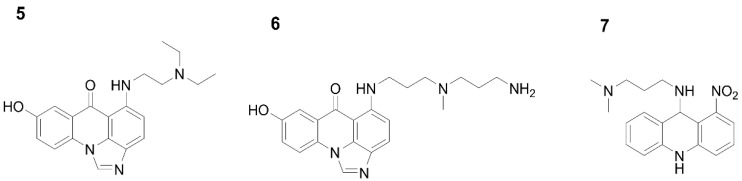
Monomers of the studied unsymmetrical bisacridine structures: 1-alkylamino-8-hydroxyimidazoacridinone (Symadex^®^, C-1311, (**5**)), 1-alkylaminoimidazoacridinone (**6**), and 1-nitro-9-alkylaminoacridine (Ledakrin, C-283, (**7**)).

**Figure 3 molecules-27-03995-f003:**
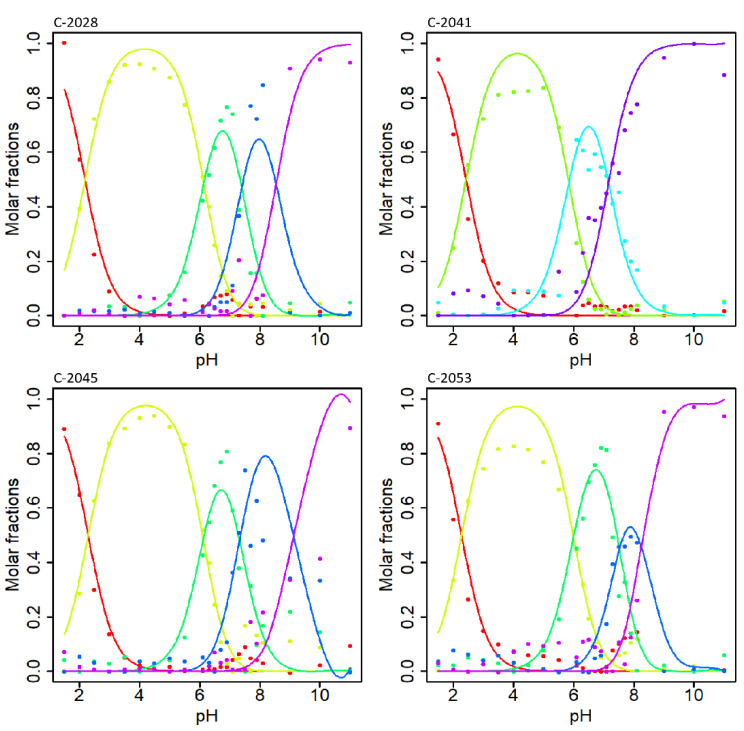
Dependence of UA molar fractions on pH values for each protonated form (points—optimal molar fractions; curves—the best fit of the theoretical model to the data). Each color corresponds to a unique spectral (i.e., protonation) form of a given molecule. The spectral forms possible for studied compounds are presented in Figure 5.

**Figure 4 molecules-27-03995-f004:**
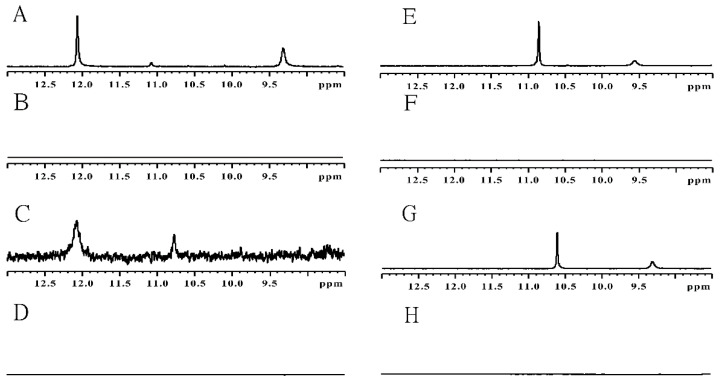
^1^H NMR spectra of UAs: C-2028 at pH = 1 (**A**) and pH = 7 (**B**), C-2041 at pH = 1 (**C**) and pH = 7 (**D**), C-2045 at pH = 1 (**E**) and pH = 7 (**F**), C-2053 at pH = 1 (**G**) and pH = 7 (**H**). Temperature = 5 °C; H_2_O/D_2_O collected in 10^−2^ M cacodylate buffer.

**Figure 5 molecules-27-03995-f005:**
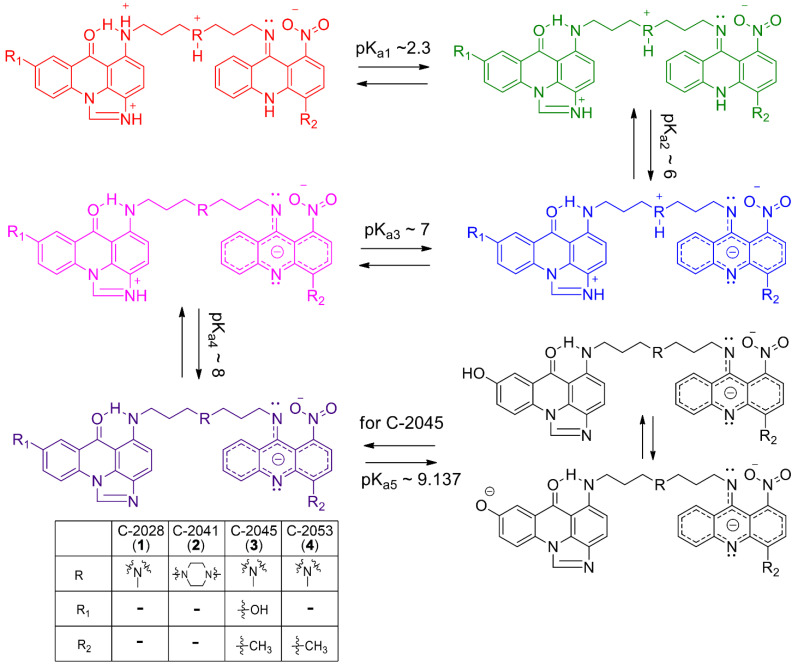
Proposed structures of individual pH-dependent forms of studied UAs (**1**–**4**) and the corresponding approximated pK_a_ values. C–2045 (**3**, black structures) was used as an example for the last pK_a_ = 9.137.

**Figure 6 molecules-27-03995-f006:**
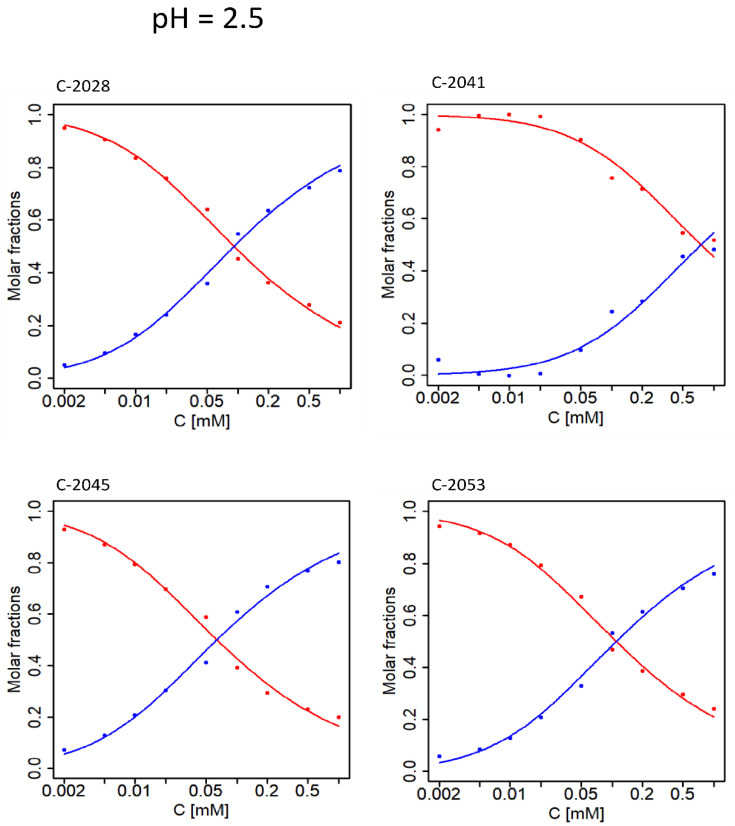
UA molar fractions for UAs studied at pH = 2.5 and 35 °C. Points—optimal molar fractions; curves—the best fit of the theoretical model of dimerization to the data. Red color—monomer, blue color—dimer.

**Figure 7 molecules-27-03995-f007:**
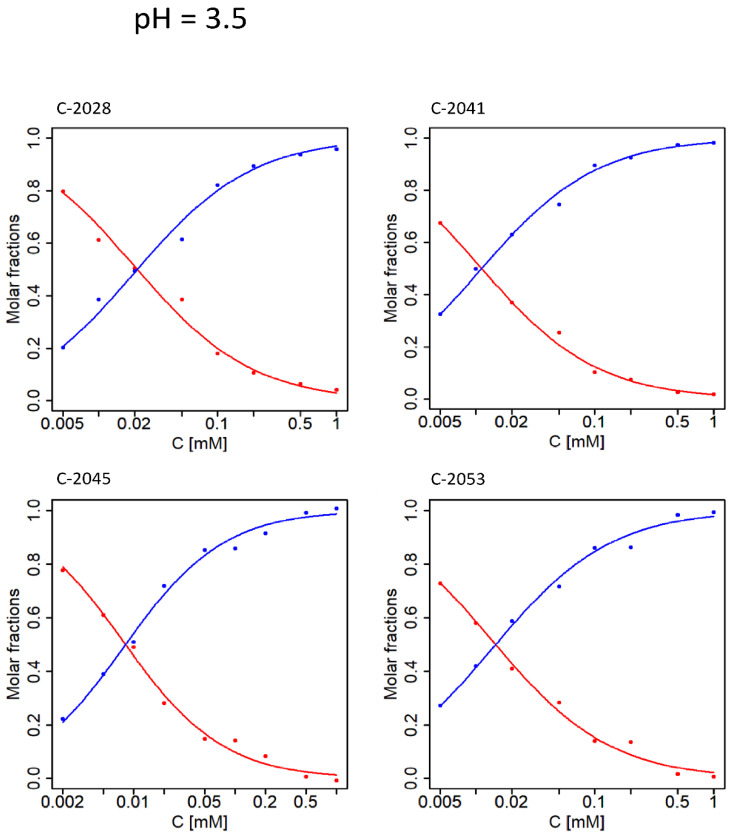
Molar fractions for UAs studied at pH = 3.5 and 35 °C. Points—optimal molar fractions; curves—the best fit of the theoretical model of dimerization to the data. Red color—monomer, blue color—dimer.

**Figure 8 molecules-27-03995-f008:**
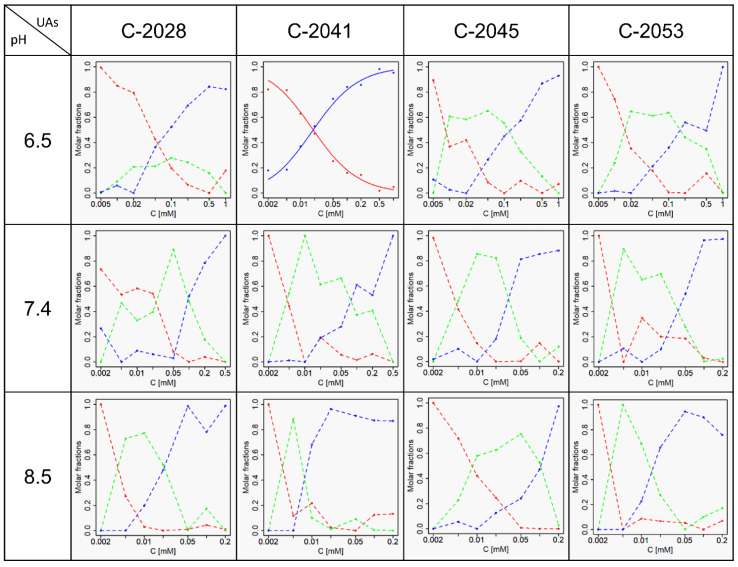
Molar fractions of UAs at pH = 6.5, 7.4, 8.5 at 35 °C. Dotted curves—molar fractions of individual self-associates, solid curves—the best fit of the theoretical model. Red color—monomer, green color—first order aggregate (presumably a dimer), blue color—higher order aggregate.

**Table 1 molecules-27-03995-t001:** pK_a_ and standard deviation values, calculated using leave-one-out cross validation.

No.	UAs	pK_a1_	SD	pK_a2_	SD	pK_a3_	SD	pK_a4_	SD	pK_a5_	SD
**1**	C-2028	2.194	0.005	6.139	0.001	7.397	0.005	8.546	0.02	-	-
**2**	C-2041	2.435	0.004	5.843	0.005	7.158	0.004	-	-	-	-
**3**	C-2045	2.301	0.003	6.120	0.002	7.327	0.006	-	-	9.137	0.04
**4**	C-2053	2.285	0.005	5.997	0.002	7.522	0.003	8.235	0.004	-	-

**Table 2 molecules-27-03995-t002:** pK_a_ and standard deviation values determined for the selected monomer structures presented in Figure 2.

No.	pK_a1_	SD	pK_a2_	SD	pK_a3_	SD
**5**	2.768	0.004	7.597	0.007	9.675	0.02
**6**	2.662	0.007	7.686	0.009	-	-
**7**	-	-	6.227	0.0007	-	-

**Table 3 molecules-27-03995-t003:** Self-association constants (K) with SD at various pH values at 35 °C. Constants marked by ‘*’ were determined via leave-one-out cross validation.

No.	pH		2.5		3.5		6.5		7.4		8.5
	UAs	K	SD	K	SD	K	SD	K	SD	K	SD
**1**	C-2028	10.89 *	0.02	26.94 *	0.033	~20		~40, ~10		~220, ~50	
**2**	C-2041	1.33	0.04	50.77 *	0.02	33.25 *	0.037	~200, ~13		~250, ~143	
**3**	C-2045	15.72 *	0.026	~63		~125, ~40		~200, ~40		~125, ~10	
**4**	C-2053	9.14 *	0.026	~50		~60, ~5		~250, ~20		~400, ~60	

## Data Availability

All data generated or analysed during this study are included in this published article and its supplementary material files.

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
