# Peer review of "Acid–Base Equilibrium and Self-Association in Relation to High Antitumor Activity of Selected Unsymmetrical Bisacridines Established by Extensive Chemometric Analysis"

_molecules, 2022, doi:10.3390/molecules27133995_

Round 1

Reviewer 1 Report

The manuscripot  

Acid-base equilibrium and self-association in relation to high 2 antitumor activity of selected unsymmetrical bisacridines es-3 tablished by extensive chemometric analysis

 reports an interesting biochemical problem. The article is well written and tha data are well presented. I am recommending to publish the manuscript after some minor corrections.

  1. It would be good to put some numbering of the structures in Fig. 1 or Fig. 5 because otherwise the reader should check the Supplementary material.
  2. P. 5, L.160 – only 1D TOCSY and ROESY are mentioned although they are also 2D
  3. Suppl. - P.6 L. 173 of the 1-nitro – a space is missing

Author Response

We thank the Reviewer for all the comments and suggestions.

  1. Figure 5, as well as its caption, has been slightly modified in order to clarify the structure numbering.
  2. This issue has been clarified in the main text.
  3. The space was added.

Also, many minor tweaks and corrections were introduced in order to improve the general accessibility of the text.

Reviewer 2 Report

The authors of manuscript molecules-1779890 describe a continuation of their studies regarding the self-association and acid-base equilibrium of unsymmetrical bisacridines. Specifically, it was to investigate how pH values relevant to normal and tumor tissue may affect the physicochemical of these anticancer compounds. I find this study perfectly suitable for publication in Molecules.

Before publication, the authors should address the following minor technical points:

1. p3, line 103: “Materials and Methods”

2. p3, line 106: Can you explain better the Nelder-Mead method using the Henderson-Hasselbalch equation?

3. p4, line 125:. You can indicate that in Figure 3 where compounds 5, 6 and 7 appear. It is not clear how you got the results from Table 2.

4. p5, line 160: Because you have performed the spectra at 5ºC if in the supporting information the NMR spectra have been performed at pH =4.5 and 25 ºC.

5. p5, line 166: “1H NMR”

6. p5, line 173: “of the 1-nitro-9-aminoacridine ”

7. p6, line 179: “Figure 4. 1H NMR…

Author Response

We thank the Reviewer for all the valuable comments and suggestions.

  1. This error has been corrected.
  2. We have added an extra reference, discussing the Nelder-Mead alghoritm. Also, we have briefly discussed this method in the Supplementary Materials and highlighted this fact in the main text.
  3. We have corrected the erroneous reference to Figure 3 instead of Figure 2. Also, the caption of Table 2 has been corrected.
  4. We have added an extra section, clarifying the usage of NMR spectra. There were, in fact, two series of experiments, recorded in 5C and 25C and this fact is now much better explained in the manuscript.
  5. This error has been corrected.
  6. The space has been addded.
  7. This caption has been corrected.

Additionaly, we have introduced numerous improvements into the main text, in order to improve its general accessibility. The language was also corrected in many places.